# Seascape Configuration and Fine-Scale Habitat Complexity Shape Parrotfish Distribution and Function across a Coral Reef Lagoon

**Maria Eggertsen** [1,*], **Dinorah H Chacin** [2,3,*], **Joshua van Lier** [4], **Linda Eggertsen** [5], **Christopher J Fulton** [4], **Shaun Wilson** [6,7], **Christina Halling** [1] and **Charlotte Berkström** [8]

1   Department of Ecology, Environment & Plant Sciences, Stockholm University, 10691 Stockholm, Sweden; christina.halling@su.se
2   College of Marine Science, University of South Florida, 140 7th Avenue South, St. Petersburg, FL 33701, USA
3   Gulf Shellfish Institute, 1905 Intermodal Circle, Suite 330, Palmetto, FL 34221, USA
4   Research School of Biology, The Australian National University, Canberra ACT 2601, Australia; josh.vanlier@gw.govt.nz (J.v.L.); christopher.fulton@anu.edu.au (C.J.F.)
5   Centro de Formação em Ciências Ambientais, Universidade Federal do Sul da Bahia, Campus Sosígenes Costa, Porto Seguro 45810-000, Brazil; eggertsen.linda@gmail.com
6   Oceans Institute, University of Western Australia, Crawley WA 6009, Australia; shaun.wilson@dbca.wa.gov.au
7   Marine Science Program, Department of Biodiversity, Conservation & Attractions, Government of Western Australia, Kensington WA 6151, Australia
8   Department of Aquatic Resources, Institute of Coastal Research, Swedish University of Agricultural Sciences, Skolgatan 6, SE-742 42 Öregrund, Sweden; charlotte.berkstrom@su.se
*   Correspondence: maria.eggertsen@su.se (M.E.); dchacin@usf.edu (D.H.C.)

**Abstract:** Structural complexity spanning fine to broad spatial scales can influence the distribution and activity of key organisms within marine ecosystems. However, the relative importance of hard (e.g., corals) and/or soft (e.g., macroalgae) structural complexity for marine organisms is often unclear. This study shows how both broad-scale (seascape configuration of coral structure) and fine-scale habitat complexity (structure height, number of holes, and presence of macroalgae) can influence the abundance and spatial ecology of reef fish. Underwater visual census of fish, surveys of habitats, remote underwater videos, and behavioral observations by following individual fish were used to quantify fine-scale habitat characteristics (e.g., complexity, coral structure height, macroalgae presence) and the abundance, size structure, and behavior (rates of herbivory, tortuosity ratios and total distance travelled) of abundant parrotfish. Both seascape configuration and macroalgae influenced the patterns of fish abundance and rates of herbivory. However, these relationships varied with trophic groups and ontogenetic stages. Abundance of adult and intermediate-phase parrotfishes was positively influenced by densely aggregated coral structures, whereas juvenile abundance was positively influenced by the presence of macroalgae. Foraging path and bite rates of an abundant parrotfish, *Chlorurus spilurus*, were not influenced by coral structure configuration or height, but the presence of macroalgae increased the bite rates of all juvenile parrotfish. Our results suggest that a combination of seascape configuration, fine-scale habitat complexity, and microhabitat selectivity influence reef fish community structure and foraging behavior, thus altering herbivory. However, these relationships can differ among functional groups of fish and life-history stages. Information on these fish–habitat interactions is critical for identifying habitats that facilitate ecological functions and ensures the successful management and conservation of essential habitats.

**Keywords:** herbivorous fish; functional diversity; ecosystem function

## 1. Introduction

Landscape configuration and habitat heterogeneity can influence both distribution patterns of organisms and how ecological processes operate over fine to broad spatial scales [1,2]. By understanding the relationships between these patterns and processes we can predict how changes within the landscape (e.g., habitat loss and fragmentation) might alter biodiversity and ecosystem function [3]. The relative importance of key features of a landscape can vary with the focal organism. For example, species distribution of birds, moths, and butterflies are mainly affected by landscape heterogeneity, while richness of amphibians and reptiles are more closely related to the abundance of specific types of habitats [4]. Habitat heterogeneity can be more important than just the presence of a certain habitat type. Loss of complexity and specific niches during the replacement of old growth forest with palm oil plantations for instance, have had devastating effects on the abundance and biodiversity of wildlife, despite the habitat type (forest) still being present [5]. Furthermore, habitat preferences are species-specific, and what might be beneficial for one species might not be advantageous for others [6]. Consequently, organisms tend to have different distribution patterns across broad- (e.g., landscape configurations) and fine-scales (habitat heterogeneity [7,8]).

Similar to terrestrial environments, correlations between seascape configuration and patterns of biodiversity are increasingly evident [9–12]. A key seascape and habitat component is benthic structural complexity, which plays an important role in maintaining high fish abundance, composition, and diversity in both temperate and tropical systems [13–19]. While the body of literature supporting these relationships is increasing in the marine realm [17], there is a lack of understanding of the mechanisms that allows us to tie patterns to ecological processes and the spatial scales on which these operate. For example, habitat heterogeneity may influence ecological processes, such as predation and herbivory [20,21], as the benthic structural complexity provides refuge from abiotic stressors, reduces predation risk, and enhances resource availability [15]. In tropical seascapes, loss of structural complexity could therefore negatively impact reef fish assemblages, and ultimately, small-scale fisheries [18,22,23].

Structural complexity can be constituted by either hard (e.g., dead or live scleractinian corals or rocks) or soft (e.g., sponges, macroalgae and/or seagrasses) structures [24,25]. The combination of hard and soft complexity might further influence fish abundances depending on species and ontogenetic stage [26–28]. In certain seascape settings, and for certain species, the effect of complexity (such as reef relief, number of holes, and seagrass or macroalgae canopy height and cover) on fish abundance can be stronger than the effect of habitat patch size, underlining the importance of this variable as an indicator of habitat quality [19,29–31]. Additionally, complexity effects at multiple spatial scales are likely to vary with the body size of fish [32–35]. Indeed, having higher complexity across different scales increases the total habitat area for a range of associated species of varying shape and size, although the relative importance of hard and/or soft structures is often unclear.

Here, we studied the coupling between broad- (i.e., seascape configuration) and fine- scale (i.e., habitat heterogeneity) complexity by addressing the following questions: How does complexity measured at different spatial scales influence the distribution and abundance of (1) fish from different functional groups and (2) scraping/excavating parrotfish at different life history stages, and (3) foraging and feeding patterns of scraping/excavating parrotfish. We focused on a backreef area/tropical lagoon where two broad spatial configurations of hard coral structure (dense or sparse) were present. Each hard structure, hereafter called "bommie," consisted of a large isolated piece of consolidated reef matrix with coral colonies growing on it [36]. These bommies were heterogeneous as they varied in height, structural complexity (number and length of cavities and bumps), and macroalgae presence. We present an overview of (1) the resident fish assemblage and (2) the most abundant group—scraping/excavating parrotfish, with respect to complexity at broad- and fine-scales. Finally, we studied foraging of scraping/excavating parrotfish within fine-scale complexity categories. For the most abundant parrotfish species (*Chlorurus sordidus*) we observed swimming behavior and calculated the distances related to seascape configuration.

## 2. Methods

### 2.1. Study Site

The study was conducted on the north coast of the island of Mo'orea (17°30′ S, 149°50′ W), Society Archipelago, French Polynesia (Figure 1). Mo'orea is a high island of volcanic origin, with surrounding barrier reefs on all sides, creating a shallow sandy lagoon with scattered coral bommies and fringing reefs (mainly *Porites* spp.). Within the lagoon, bommies are arranged in both sparse and high densities [37,38]. A number of channels connect the lagoon with the open ocean through a total of 12 passes intersecting the barrier reef [37]. Deeper areas (~30 m) in the north part of the lagoon are found in the two bays, which are characterized by a siltier substrate [37]. Tidal range is small (~ 0.3 m), and water exchange within the lagoon is mainly driven by waves breaking over the barrier reef [39,40]. Since the 1980s, the reef communities of Mo'orea have been subjected to repeated coral bleaching events, several cyclones and *Acanthaster* spp. (COTS) outbreaks, which have reduced coral cover and shifted composition from *Acropora-* to a more *Porites*-dominated community [41,42]. In certain areas, the dead coral substrate on the bommies is frequently colonized by the brown macroalgae *Sargassum* spp. and the introduced *Turbinaria ornata* [43–46].

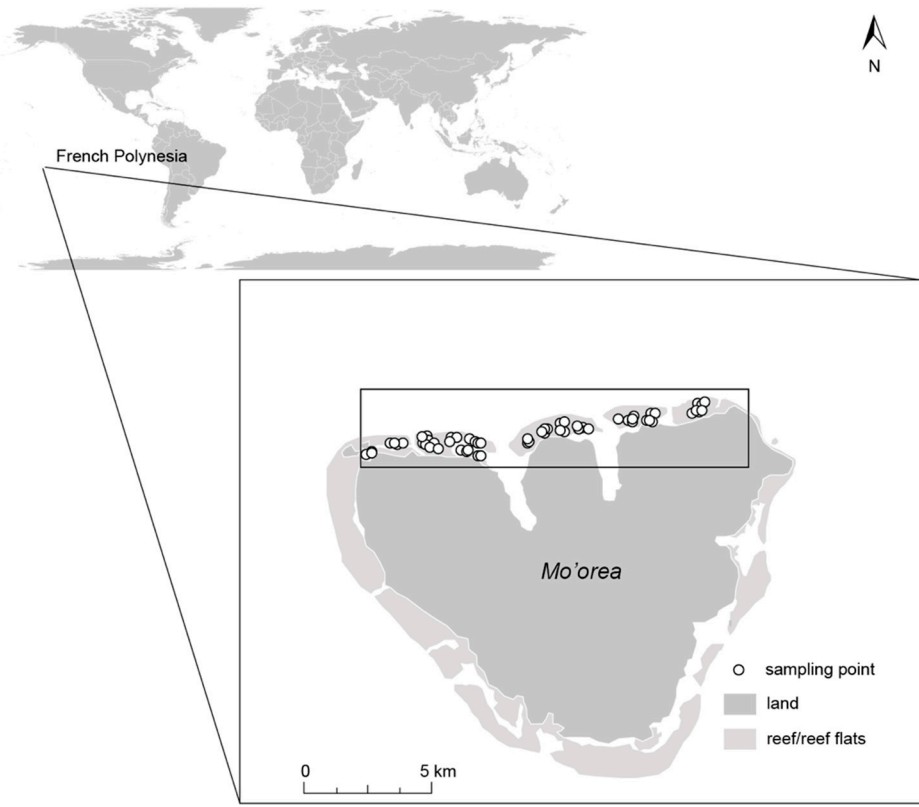

**Figure 1.** The island of Mo'orea, French Polynesia and its location in the Pacific Ocean. The north shore, where all survey points were located, is indicated with a black square.

### 2.2. Survey Design and Data Collection

#### 2.2.1. Seascape and Habitat Categorization

Field surveys were conducted along the northern shore of Mo'orea (Figure 1). All surveys were performed within the lagoon (the back-reef habitat) at 1–3 m depth during October 2017. Our survey points consisted of a two-part process. Part 1 consisted of selecting the location for the survey point (based on a series of seascape and habitat criteria) and part 2 consisted of documenting the fish assemblage and finer-scale habitat characteristics (see Section 2.2.2 Fish and habitat survey) in

the chosen location. The three criteria included in part 1 were: (1) configuration of coral bommies, (2) height of coral bommies, and (3) presence/absence of macroalgae. Seascape configuration reflected densely or sparsely aggregated coral bommies (Figure 2a; Supplementary Materials Tables S1 and S2). Survey points were chosen from satellite imagery where the two types of seascape configurations could be easily identified. These locations were further classified as "tall" if the bommies' height was >0.5 m or "short" if the bommies were ≤0.5 m in height (Figure 2b). Additionally, bommies with macroalgae (*Padina boryana*, *Sargassum* spp. and *Turbinaria ornata*) were classified as "macroalgae present," while bommies without macroalgae were classified as "macroalgae absent" (Figure 2c). All of these habitat classifications were present in both dense and sparse seascape configurations. This site selection, based on the criteria just mentioned, resulted in 42 survey points in dense configurations comprising 22 survey points on short bommies and 20 on tall bommies; while 44 point surveys were conducted on sparse configurations comprising 22 surveys on short bommies and 22 on tall bommies (Figure S1). Care was taken to choose survey points that were considerably large (>50 m across) and consistent with the category assigned.

### 2.2.2. Fish and Habitat Surveys

All surveys were conducted between 09:00 and 16:00 to avoid any behavioral differences associated with sunrise or sunset [47]. This survey period also covered the most active time for herbivorous fish [48], which were the focus of the present study. Once the location was selected according to the above explained method, fish abundances and finer-scale habitat variables were estimated using a standardized point count method following Berkström et al. [49]. To minimize any variation related to the number of observers all point count surveys were performed by ME and DHC. Prior to the point count surveys, practice of fish lengths estimations was conducted to minimize size estimate errors and to calibrate both observers [28]. Forty-two point count surveys were carried out in dense habitats and 44 in sparse habitats (Figure 2, Figure S1). Each point count survey measured 5 m in diameter and a snorkeler was positioned within the circle documenting all roving fish within the circle, identifying them to the highest taxonomical level (usually species) and estimating their total length to the closest cm [50]. After 3 min, the observer swam in a back-and-forth pattern, covering the whole circle and documented all cryptic fish species and small juveniles (species identification, numbers and length). As the observed fish were continuously counted for 3 min, it was impossible to discern if any fish entering and leaving the surveyed cylinder area was counted more than once. We assume that if this occurred, it would have occurred similarly across our census.

Within each point coint survey, the number of coral bommies, height and diameter of each bommie, cover of bommies, and other bottom substrate (% cover, mainly sand and rubble) were noted. For each point count survey, one bommie was randomly chosen for more detailed mapping. A measuring tape was placed over the widest part of the bommie and the percent cover of live coral, crustose coralline algae (CCA), epilithic algal matrix (EAM), macroalgae, and dead coral was estimated by measuring habitat cover immediately under the tape. This method has previously been used to effectively estimate the percent substrate cover in tropical marine habitats [51]. When macroalgae was present; number of holdfasts of large macroalgal species (*Sargassum* spp. and *Turbinaria ornata*), height of the 3 tallest macroalgal individuals, and percent cover of different species of macroalgae were noted. Fine-scale structural complexity of the bommie was estimated on a scale ranging from 0–5 following definitions in Polunin and Roberts [52]. This was determined by measuring cavities and bumps on the bommie structure and classifying as following: 0 = flat, 1 = <10 cm, 2 = 11–30 cm, 3 = 31–60 cm, 4 = 61–100 cm, 5 = >100 cm (Figure 2d). Within the surveyed point survey area, the sea urchin abundance was also noted. Location of each point census was marked with a GPS (Garmin etrex 10) so that the distance to the barrier reef could be calculated using the function "near" in ArcGIS 10.5.

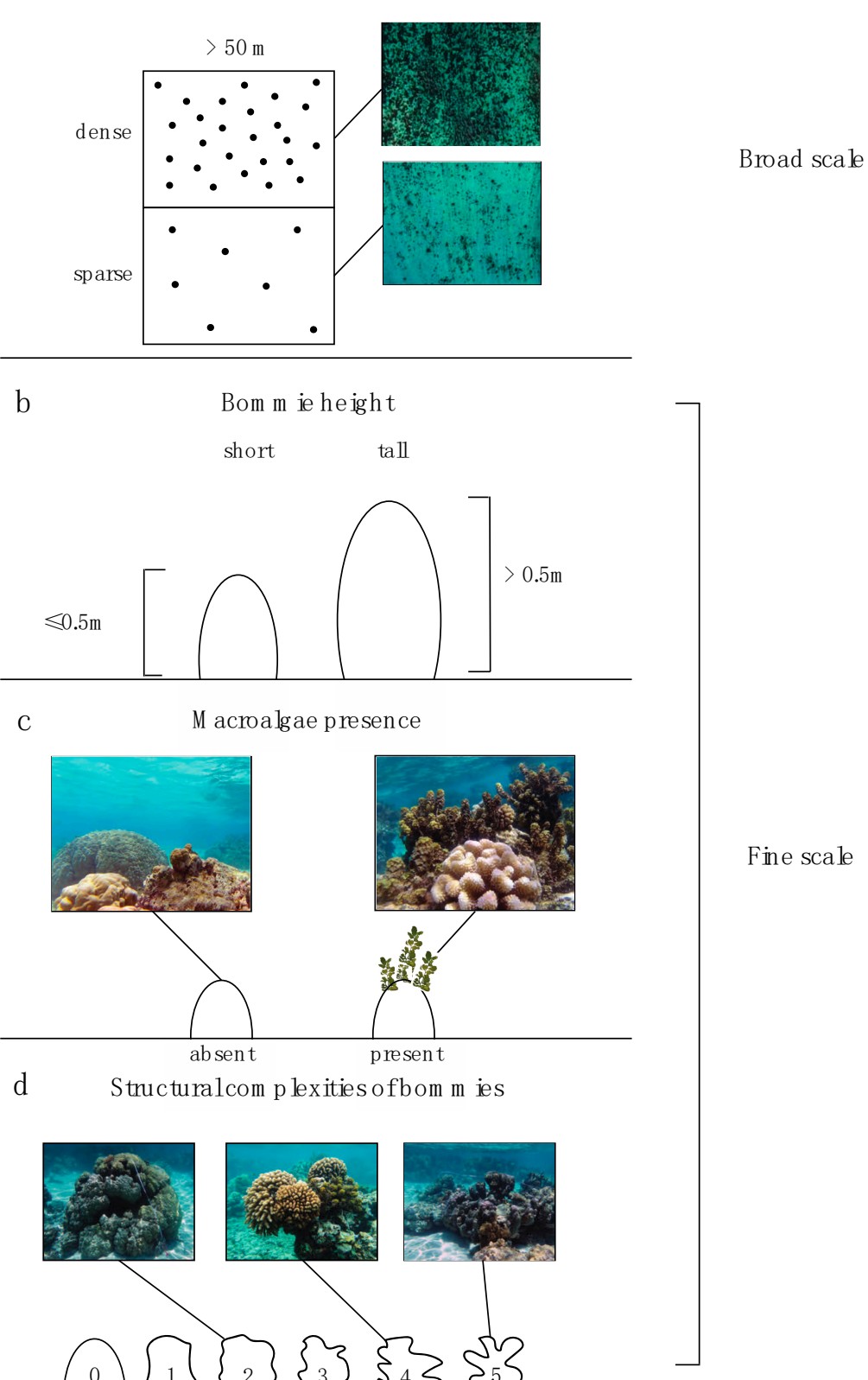

**Figure 2.** Complexity levels at multiple spatial scales; (**a**) broad scale configuration of coral bommies in the seascape ($n_{dense}$ = 42 and $n_{sparse}$ = 44), (**b**) height of bommies ($n_{short}$ = 42 and $n_{tall}$ = 44), (**c**) presence or absence of macroalgae on the bommie structure ($n_{absent}$ = 43 and $n_{present}$ = 43) (**d**) fine-scale structural complexity of bommies ($n_2$ = 5, $n_3$ = 27, $n_4$ = 22, $n_5$ = 32).

### 2.2.3. Fish Spatial Movement

Short-term spatial movements (tortuosity ratios and total distance travelled) of one of the most abundant parrotfish in Mo'orea, *Chlorurus spilurus* [42] were quantified within areas of two different hard structure configurations (dense vs. sparse) and bommie heights (tall vs. short). For detection of differences in movement patterns between hard structure configurations and height, we used total travel distance and the tortuosity of their travelling path following Fulton and Bellwood [53]. A total of 54 individuals were observed of which one individual was a terminal phase male, 44 were initial phase, and 9 were juveniles. These were distributed in dense (n = 26) and sparse (n = 28) configurations.

Individual *C. spilurus* were randomly chosen and followed by a snorkeler for 2–3 min. Each individual was followed from a distance of ~2 m to minimize any potential disturbance by the snorkeler. If fish behavior appeared to be influenced by the snorkeler (e.g., abrupt interruption of feeding, sudden fast swimming in opposite direction, unwanted attention toward the snorkeler), the session was terminated and the data excluded from analyses. In case of unnoticed disturbance, we assumed that these occurred similarly across our trials. Size of each fish (total length) was estimated to the nearest cm and social behavior was noted, i.e., if the fish was in solitary or together with other fish. Each snorkeler was equipped with a GPS (Garmin etrex 10) placed in a waterproof bag floating at the surface, positioned as close as possible to the snorkeler. The GPS was set on tracking mode allowing continuous recording of the geographical position every 10th second. Start and finish times of each observation were recorded by the snorkeler using a watch that was synchronized with the GPS. Tracks were imported into QGIS (version 2.17), where swimming distances were mapped and measured. Standardized mean values of either 2 or 3 min were used (total and linear distance) for calculating tortuosity ratios.

### 2.2.4. Fish Bite Rate

Remote underwater videos (RUV) were used to investigate differences in feeding behavior (bite rates) by scraping/excavating parrotfish among habitat categories (dense and sparse, tall and short bommies, and presence or absence of macroalgae). A GoPro camera (Hero 3/Hero 4) was mounted on top of a coral bommie, so that it recorded an area of approximately 2 m$^2$. The plot area was defined by a snorkeler at the beginning of each recording by placing a 1-m line in front of the camera and then removing it before feeding observations were recorded. Each GoPro video recorded 5–10 min. When the video was completed, the depth was noted and the location was marked with a GPS waypoint. Number and species of scraping/excavating parrotfish, number of bites, and bite substrate within the plot area were identified from the video recordings. A bite was defined as the jaws of a fish being in contact with the substratum. Bite rates were standardized per minute and mean bite rates per life stage were calculated. We conducted 92 RUVs of which 6 were discarded because of unsatisfactory video quality or field of view. Of the remaining 86 RUVs, 51 were conducted in dense habitats and 35 in sparse.

Fine-scale structural complexity (rugosity) of the videoed plot area was calculated using photogrammetry similar to Young et al. [54], by generating a 3-dimensional (3D) model of the plot from where the total surface area was calculated, and then divided by the total 2-dimensional (2D) area. The ratio was used as a proxy for rugosity and was calculated as:

$$1 - \frac{2D\ area}{3D\ area}$$

Photogrammetry is a relatively fast and non-destructive method, which has been shown to estimate fine-scale structural complexity with high accuracy [54,55]. Structural complexity can affect grazing activity among herbivorous fish [20] and was used instead of the earlier described 0–5 graded scale, as it gave a precise measure of complexity relevant to the area being videoed.

To obtain suitable imagery for photogrammetry used for measuring the effect of fine-scale structural complexity on bite rates, each plot area was filmed by a GoPro (Hero3/Hero4) camera,

following methods in Young et al. [54]. The camera was kept at a constant distance from the substrate (~0.5–1 m) by a snorkeler who swam slowly in a back-and-forth pattern, carefully covering all areas of the plot. Cameras were set on video mode, resolution 1080 p and no wide-angle. We used the software FFmpeg (www.ffmpeg.com) to extract photographs of ~80 percent overlap from the video sequences (3 frames second$^{-1}$). To generate the 3D-models, we used AgiSoft PhotoScan (version 1.4.3., Standard version, 2018 AgiSoft LLC, St. Petersburg, Russia). Model rendering followed a standard procedure and the same settings described by Young et al. [54] including aligning photos, building dense point cloud, building mesh and building texture. If models were not accurately constructed because of e.g., light distortion they were discarded and not used for further analysis. Models were exported in .obj format and imported into MeshLab [56]. The plot area in each model (defined from the GoPro videos) were identified and all other parts of the 3D-model removed. The remaining model was cleaned from graphical errors i.e., vertexes/faces disconnected from the surface of the defined area were removed. Total 3D-area was calculated using function "Filters/Quality Measure" and "Computations/Compute Geometric Measures." The 2D area was calculated using the measuring tape tool.

*2.3. Statistical Analyses*

Fish from point count surveys and RUVs were classified into seven main trophic groups (corallivores, detritivores, herbivores, invertivores, omnivores, piscivores, and planktivores) according to available literature and FishBase information [57–61]. As mobile herbivorous fish were the most abundant functional group, we decided to focus on these and further classified them into three finer categories: "browsers", "grazers", and "scraper/excavators" [57,58,60–62]. This resulted in the subgroup "scrapers/excavators" containing both scraping and excavating parrotfish, which perform different functional roles on reefs. These were merged into one group because the majority of individuals were juvenile/initial phase parrotfish and difficult to identify to species level. Scraping and excavating parrotfish from the point count surveys and spatial movement studies were classified either as juveniles or adults following Tano et al. [63] using the 1/3 method or length at maturity (L$_M$) values when available. A mean value was calculated for L$_M$ for the two most commonly observed parrotfish (*Chlorurus spilurus* and *Scarus globiceps*) [64–66]. This value was used for classifying all parrotfish because of the difficulties in identifying small individuals to species. All parrotfish with a total length (TL) less than half of the L$_M$ value (7.7 cm) were considered juveniles, and all individuals larger than that were classified as subadults/initial phase or terminal phase, if not contradicted by coloration. For parrotfish observed in the RUVs, coloration was used for life stage characterization (juvenile, initial phase or terminal phase) because size estimation with accuracy was not possible.

To test how the different broad-scale complexity categories influenced the number of total, herbivorous, and scraping/excavating parrotfish (both total numbers and separated by life stage), a three-way analysis of variance (ANOVA) was performed with bommie configuration (dense or sparse), bommie height (short or tall), and macroalgae (presence or absence) as factors. Additionally, interactions among factors were tested. Because only a few terminal phase individuals were recorded (total = 9 in all point count surveys), this group was pooled with the subadult/initial phase for analyses. To test how environmental variables influenced the response variables of fish density, subadult/adult scrapers/excavators density, and juvenile scrapers/excavators density on a more detailed level, a full subset multiple regression approach was used [67]. The environmental predictor variables included coral cover (%), sand cover (%), EAM cover (%), CCA cover (%), macroalgal cover (%), macroalgae height (cm), number of macroalgae holdfasts, number of urchins, distance to barrier reef (m), fine-scale structural complexity (1–5), depth (m), number of bommies, and height of bommies (m). A generalized additive model (GAM) with a tweedie distribution from package "mgcv" [68] was used. The tweedie distribution belongs to the family of exponential distributions together with e.g., poisson and gamma distributions, but has shown superior fits for overdispersed and zero-inflated data compared to other members of this family [69], which is why this distribution was chosen. "Depth" and "site" were set as random factors and added in all models including the null model by using the bs = "re" specification.

Maximum number of predictors was set to three. Because of limited replication regarding interacting factors, interactions were excluded from this analysis. The most parsimonious models were defined based on Aikaike Information Criterion with correction for small sample sizes (AICc), which was defined as models with $\Delta$AICc $\geq$ 2. The full subsets function automatically control for collinearity, and removes predictors that are too closely correlated (>0.28) [67,70]. Because of highly correlated predictors, it was not possible to analyze bommie configuration and the environmental variables in the same models.

The effect of bommie configuration (dense vs. sparse) and bommie height (short or tall) on spatial movements of *C. spilurus* (tortuosity ratio and swimming distance min$^{-1}$) was tested by using three-way ANOVAs with bommie configuration, bommie height and life stage as factors. Only one individual was defined as a terminal phase male in the dataset, and this individual was removed from statistical analysis, because movement patterns might differ between life stages.

The effect of bommie configuration (dense or sparse), bommie height (short or tall), distance to barrier reef (m) and 3D-complexity index on bite rates (number of bites min$^{-1}$) from the RUVs were analyzed with linear models or (when criterion for normality was not fulfilled) generalized linear models (GLMs) from package "stats" (R Core Team 2017) with poisson distribution and loglink function. Response variable "bite rates" either consisted of all herbivorous fish pooled or separated into "total detritivores", "total scraping/excavating parrotfish", "juvenile scraping/excavating parrotfish", and "adult scraping/excavating parrotfish". Juvenile detritivores were not analyzed because of small sample size. Predictor variables were checked for multicollinearity by pairwise comparison (Spearman rank test) and examination of variation inflation factor (VIF) values [71]. Predictors with VIF values of >2, were removed from the same model.

Prior to all analysis, normal distributions of predictor and response variables were checked by visual examination using diagnostic plots, and if necessary, log(x + 1) or square root transformation were applied. Criteria for normality was fulfilled for all linear models after transformation (total, herbivorous, and scraping/excavating parrotfish density in broad- and fine- scale habitats, adult and juvenile scraping/excavating parrotfish density in broad- and fine- scale habitats, and spatial movements of *C. spilurus* in broad- and fine- scale habitats).

## 3. Results

*3.1. Fish Assemblage in Different Seascape Configurations and Habitat Complexities*

A total of 2,312 individuals from 79 different species, 52 genera, and 25 families were recorded during the point count surveys (Table S3). Small-bodied herbivores were the most abundant fishes in almost all habitat categories (Figure 3).

Total fish abundance and herbivorous fish abundance were higher in dense versus sparse bommie configurations (Table 1). Additionally, total fish abundance was higher in "tall" compared to "short" bommie habitats, though there was no significant effect of macroalgae (Table 1).

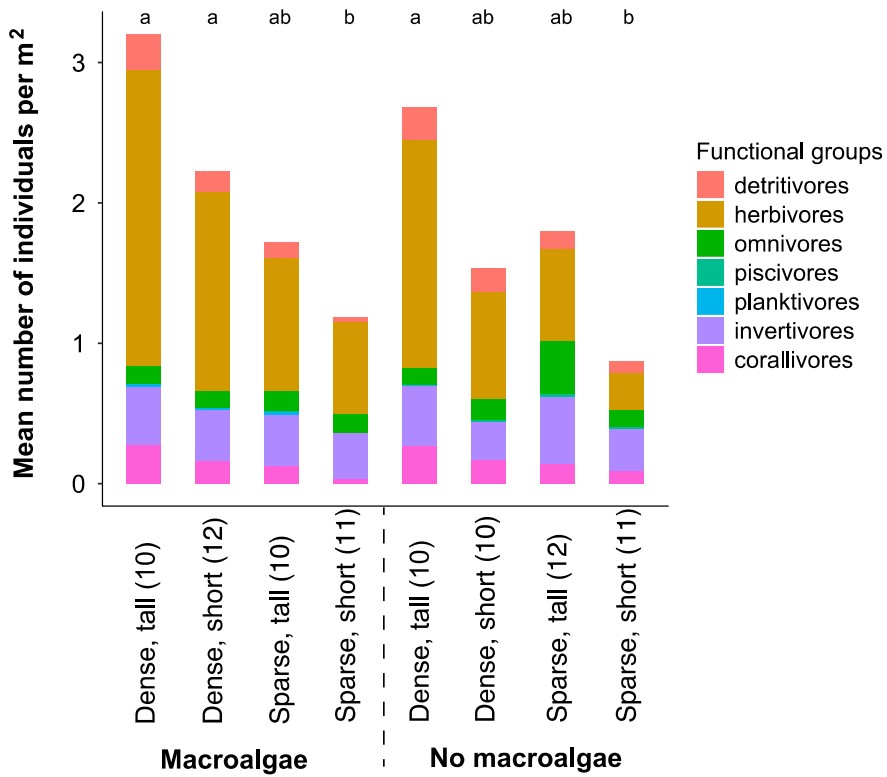

**Figure 3.** Mean fish density in different trophic groups extracted from the underwater visual census (point count surveys) surveys in the different broad- (dense and sparse configuration of bommies) and fine- scale (tall or short bommie height, macroalgae presence or absence) complexity habitats. Sample sizes are presented in parentheses (n). Letters above bar graphs that differ denote significant differences among habitat categories.

**Table 1.** Summary statistics from ANOVA models of the effects of seascape configuration (dense versus sparse bommies), bommie height (short or tall), and the presence/absence of macroalgae on the abundances of all fish and on herbivorous fish. Numbers in bold denote significant *p*-values ($p < 0.05$).

| Complexity Category | Df | MS | *F* | *p*-Value |
|---|---|---|---|---|
| **Total Fish Abundance** | | | | |
| Configuration | 1 | 3.919 | 16.161 | **<0.001** |
| Bommie height | 1 | 4.420 | 18.229 | **<0.001** |
| Macroalgal presence/absence | 1 | 0.421 | 1.735 | 0.19 |
| Residuals | 82 | 0.242 | | |
| **Herbivorous Fish Abundance** | | | | |
| Configuration | 1 | 11.473 | 11.546 | **0.0011** |
| Bommie height | 1 | 2.575 | 2.592 | 0.111 |
| Macroalgal presence/absence | 1 | 4.639 | 4.669 | **0.034** |
| Residuals | 82 | 0.994 | | |

Scraping/excavating parrotfish were the most abundant group among herbivorous fish, though densities varied among habitat categories (Figure 4) and the influence of habitat on fish abundance differed among adult and juveniles (Figure 5, Table 2).

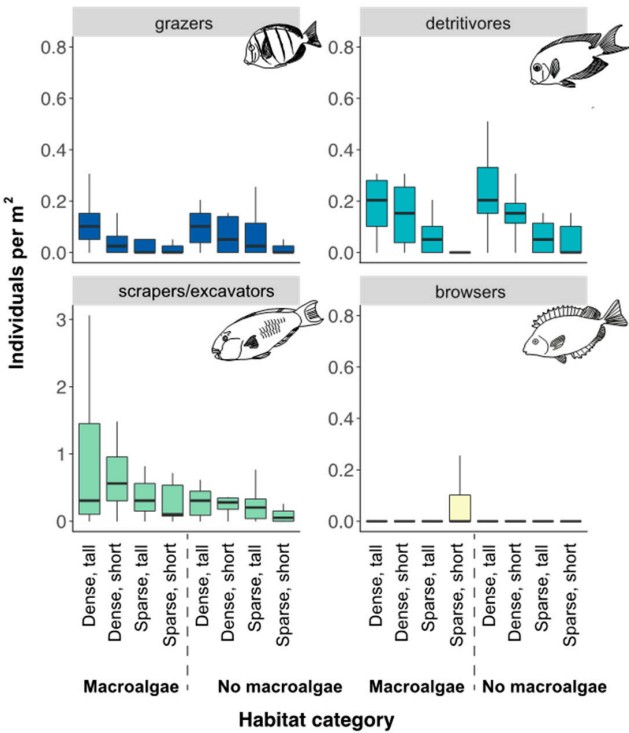

**Figure 4.** Densities of functional groups of herbivorous fish (grazers, browsers, scrapers/excavators, and detritivores) from the point count surveys. Whiskers are 95% confidence intervals of the median. Each box shows median (black line) and 25th and 75th percentile. Two outliers in the grazer group in dense-short-no macroalgae habitat (caused by a school of 74 and 75 individuals of *Acanthurus triostegus*) were removed from the figure for better visualization of plots. Note, different values on scraper/excavator x-axis compared to the other functional groups.

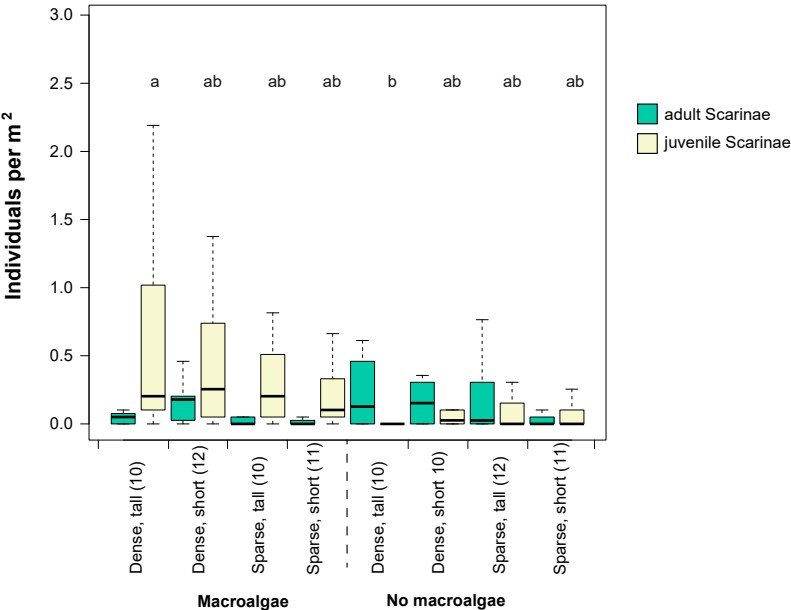

**Figure 5.** Densities of size classes of scraping/excavating parrotfish in different seascape configurations and habitat complexity categories from the point count surveys. Whiskers are 95% confidence intervals of the median. Each box shows median (black line) and 25th and 75th percentile. Letters above bar graphs that differ denote significant differences among habitat categories. Numbers in brackets denote sample size (n) in that particular habitat category.

**Table 2.** Summary statistics of ANOVA models of the effects of seascape configurations (dense versus sparse bommies), bommie height (short or high), and presence/absence of macroalgae on the abundance of scraping/excavating parrotfish at different ontogenetic stages (adults or juveniles). Numbers in bold indicate significant *p*-values ($p < 0.05$).

| Complexity Category | Df | MS | F | *p*-Value |
|---|---|---|---|---|
| **Adult Parrotfish Abundance** | | | | |
| Configuration | 1 | 7.975 | 9.770 | **0.002** |
| Bommie height | 1 | 0.011 | 0.014 | 0.906 |
| Macroalgal presence/absence | 1 | 0.673 | 0.824 | 0.366 |
| Residuals | 81 | 0.816 | | |
| **Juvenile Parrotfish Abundance** | | | | |
| Configuration | 1 | 0.728 | 0.726 | 0.397 |
| Bommie height | 1 | 0.424 | 0.423 | 0.517 |
| Macroalgal presence/absence | 1 | 23.613 | 23.548 | **<0.001** |
| Residuals | 81 | 1.003 | | |

Adult scraping/excavating parrotfish were more abundant in dense versus sparse seascapes, whereas no differences were evident among areas with short vs. tall bommies (Table 2, Figure 5). In contrast, juvenile parrotfish were significantly more abundant in habitats with macroalgae present and were not significantly affected by bommie configuration or height (Table 2). Interactions among all factors tested were not significant for either total fish abundance, herbivorous fish abundance, or total parrotfish abundance (juveniles and adults; Tables S4 and S5).

*3.2. Environmental Variables Influencing Total Fish and Scraping/Excavating Parrotfish Abundance*

Total fish abundance was positively influenced by bommie height, macroalgal cover on bommies, and turf cover, and negatively influenced by sand cover and distance to barrier reef (Table 3, Figure S2). Notably, bommie height was in both of the top models suggesting it has a strong influence of total fish abundance. Macroalgal density (number of holdfasts), distance to barrier reef, and number of sea urchins were the best predictor variables for juveniles of scraping/excavating parrotfish (Table 3, Figure S2) and had a positive influence on these fish. Holdfast density was especially important occurring in all of the top models. Conversely, no macroalgal predictors were important for the subadult/adult parrotfish, whose abundance was best predicted by distance to barrier reef, sand cover, bommie height, and coral cover which all had a negative influence. Number of bommies had a positive influence for this group.

**Table 3.** Most parsimonious models explaining spatial variation in total fish and parrotfish (adult and juvenile) abundance. Results based on generalized additive model (GAM) and full subset models. All models with ΔAICc < 2 are shown.

| Response | Model | ΔAICc | AICc Weights | R² | Edf |
|---|---|---|---|---|---|
| Total fish abundance | Bommie height + log(Turf cover) + Sand cover | 0 | 0.478 | 0.233 | 6.86 |
| Total fish abundance | Bommie height + log(Distance to barrier reef) + sqrt(Macroalgal cover on bommie) | 0.604 | 0.353 | 0.235 | 6.56 |
| Juvenile parrotfish abundance | log(Number of holdfasts) | 0 | 0.056 | 0.311 | 11.82 |
| Juvenile parrotfish abundance | log(Distance to barrier reef) + log(Number of holdfasts) | 0.94 | 0.036 | 0.331 | 12.64 |
| Juvenile parrotfish abundance | log(Number of holdfasts) + Number of urchins | 0.95 | 0.035 | 0.314 | 12.9 |
| Adult parrotfish abundance | Bommie height + Sand cover | 0 | 0.086 | 0.145 | 12.26 |
| Adult parrotfish abundance | log(Distance to barrier reef) + Number of bommies + sqrt(Coral cover) | 0.022 | 0.085 | 0.176 | 11.33 |
| Adult parrotfish abundance | Bommie height + Sand cover + sqrt(Coral cover) | 0.071 | 0.083 | 0.161 | 12.63 |
| Adult parrotfish abundance | Sand cover | 1.17 | 0.048 | 0.127 | 10.53 |
| Adult parrotfish abundance | log(Distance to barrier reef) + sqrt(Coral cover) | 1.813 | 0.035 | 0.135 | 9.02 |

*3.3. Spatial Movements of Scraping/Excavating Parrotfish in Different Seascape Configurations and Habitat Complexities*

Tortuosity ratios of both adult and juvenile parrotfish did not differ significantly between dense and sparse bommie configurations or between habitats with tall or short bommies. Mean distance travelled per min did, however differ significantly between life stages of *C. spilurus* (Table S6, ANOVA, F = 6.652, df = 1, *p* = 0.036), regardless of seascape configuration (Table S6, ANOVA, F = 2.457, df = 1, *p* = 0.123), with juveniles moving considerably less than adult/initial phase individuals (3.57 ± 1.19 m per min compared with 7.82 ± 1.18 m per min).

*3.4. Bite Rates of Herbivorous Fish and Seascape Configuration*

Total number of bites $min^{-1}$ of all herbivores and detritivores pooled was higher in dense than in sparse bommie configurations (mixed linear model, F = 6.073, df = 1, *p* = 0.016). However, when bite rates were standardized per individual, there was no significant effect of seascape configuration on bite rates.

Furthermore, none of the other tested variables had any effect on bite rates per individual (bommie configuration, bommie height, rugosity, presence/absence of macroalgae or distance to barrier reef) for all roving herbivores and detritivores pooled, for herbivores and detritivores analyzed separately, herbivores separated according to functional groups (grazers or scrapers/excavators), adult fish, or families of fish (Acanthuridae and Scarinae). There were no browsers detected in the RUVs and hence effects on these could not be tested. Additionally, no effects on bite rates were detected for initial/terminal phased parrotfish, but the presence of macroalgae (*T. ornata* or *Sargassum* spp.) increased the bite rates of individual juvenile parrotfish (GLM, *z*-value = 2.675, *p* = 0.007).

Rugosity calculated by photogrammetry in RUVs had a significant positive effect on the total fish abundance (linear regression model, F = 6.091, df = 1, *p* = 0.016).

## 4. Discussion

Both configuration of hard coral structure (dense vs. sparse bommie density) and the presence of macroalgae growing on the coral structure influenced reef fish abundance and herbivory. However, these relationships did not apply to the whole fish community, and differed among trophic groups of fish and life history stages. The amount and height of hard structure, the distance to the barrier reef, coral, and sand cover were important for predicting abundance of adult scraping/excavating parrotfish, but had little effect on the abundance of juveniles. Number of juvenile parrotfish was instead positively related to the density and presence of macroalgae growing on the hard bommie structures. This study showed that complexity measures at multiple spatial scales influenced marine fish, but that their relative importance can vary with life stage. Moreover, this study elucidated how hard complexity (coral structure) and soft complexity (canopy-forming macroalgae) can simultaneously influence herbivory. Each of these complexity categories has been evaluated in previous studies and are known for being ecologically relevant, but few studies have investigated the role that both factors may play for the abundance and distribution of marine fish. Herbivory has been stated as a key function for reef resilience [72–74], and studies that increase our knowledge of ecologically relevant factors that may affect herbivory and distribution and abundance of herbivorous fish are of essence to continue developing better management and conservation plans. This study showed that macroalgae was particularly important for juvenile parrotfish, but further information is needed on how other herbivores might respond to changes in canopy-forming macroalgal abundance and the mechanisms driving these relationships.

Densely configured bommies were grazed to a significantly higher degree than sparse bommies, which is consistent with other studies where herbivory is positively related to structural complexity [20,75]. However, this relationship, in the current study, was explained by the higher number of scrapers/excavators observed in densely configured bommie habitats as there was no effect on bite rates per individual fish. A higher amount of hard structure alone might therefore not

induce additive effects in the form of non-linear increases in ecological processes (i.e., herbivory), but will increase herbivory because this habitat type is more attractive to certain functional groups of fish than habitats with less hard structure. Likewise, our study could not detect any differences in foraging behavior of the parrotfish *Chlorurus spilurus* according to configuration of hard structure (dense vs. sparse) or height of bommies (tall vs. short), indicating that individuals did not spend more energy on swimming in particular habitat categories. However, there was a difference in distance travelled between life stages of *C. spilurus* as adults travelled longer distances compared to juveniles, which resided in relatively small areas.

Some aspects of fine-scale habitat complexity (i.e., macroalgae presence and density) were important for certain groups of fish indicating a positive effect of added soft structure. Macroalgae (*Sargassum* spp. and *Turbinaria ornata*) holdfast density for example was positively related to the abundance of juvenile scraping/excavating parrotfish, but macroalgae had negligible influence on the abundance of adult parrotfish. Contrasting patterns between different life-stages of parrotfish have similarly been observed in the Western Indian Ocean [28], suggesting that macroalgae in shallow areas constitute a beneficial habitat for some juvenile parrotfish species. Indeed, this has been the case in juvenile *Leptoscarus vaigiensis,* which are associated with high-density canopy-forming macroalgae [76]. Pockets of canopy-forming macroalgae embedded within the coral reef matrix might be important for the recruitment of some species including taxa important to local fisheries [77].

Likewise, abundance of *C. spilurus,* a prominent scarid in Mo'orea and other locations [42,78,79] responds positively to reef disturbances that reduce coral cover, but where structural complexity is still retained [80,81]. Scraping functions may therefore be preserved in areas subjected to loss of live coral [22]. However, the extent to which scarids and other ecologically relevant species depend on the availability of macroalgal nursery habitats (e.g., macroalgal patches) or increased food resources due to larger areas of feeding substrate, needs further investigation [82].

Our finding that juvenile scraping/excavating parrotfish took significantly more bites when macroalgae was present infers that macroalgae is important habitat for these fish. A possible explanation for this association is that macroalgae in combination with hard structure might provide both shelter and food for some species of juvenile parrotfish, allowing them to safely feed during longer periods. Importantly, no juvenile parrotfish were observed removing tissue from either *Sargassum* spp. or *T. ornata* "leafs," instead they were likely targeting epiphytes, biofilm, or small invertebrates living on the macroalgal thallus [83,84]. Conversely, older and larger parrotfish, such as *C. spilurus*, feed from the benthic substrate by excavating bites. Areas with a high degree of macroalgae might therefore not be attractive to larger individuals. Further studies are however required to confirm the dietary sources of these fish and how their ecological functions vary with body size.

Structural complexity of hard substrate is widely known to enhance reef fish abundance and diversity [16,17,22], but very few have studied the combined effects of hard and soft structure (e.g., macroalgae) on fish assemblage structure [19] or key ecosystem processes (but see Fulton et al. [85]). There are various mechanisms explaining how increased complexity might influence fish assemblages. First, higher hard structural complexity increases surface area, which can sustain a greater density and variety of resources such as invertebrates and microhabitats for algae [38,85,86]. Second, complexity provided by macroalgae may be important for epibionts and the distribution of fish that feed upon them [51]. Third, higher structural complexity can offer refuges for fish from predators [87]. Surprisingly we could not detect an effect of hard fine-scale complexity (scale 0–5) on fish abundance in the point count surveys, which could have been influenced by the low number of replicates in the different categories. This was further supported by the results from our rugosity calculations using photogrammetry, where rugosity did have an effect on fish abundance. Hard structure may also have an important role during dark hours in providing shelter and sleeping grounds for several fish, including parrotfish [32,88], but this warrants further investigation. Similarly, the role of hard structure during night time may differ in other fish functional groups such as invertivores as some may follow invertebrate diel peak activity and/or move to other habitats for feeding [47,89].

Structural complexity at finer spatial scales has also been shown to influence herbivory. For example, small crevices and holes can prevent herbivores from reaching algae growing within the crevices and thus constitute microtopographic refuges [38,90]. Some larger herbivores (e.g., adult parrotfish) might not efficiently forage on high structural complex surfaces, and may prefer to target less complex and more exposed surfaces [86]. This illustrates that habitat-species functional relationships are a complex interplay between different factors at both fine and broad scales, highlighting the importance of considering complexity across multiple spatial scales.

Our study also highlights the need to take ontogeny into account when evaluating the influence of habitat characteristics on fish assemblages, and their functional roles in tropical seascapes. Most of the parrotfish observed in our study were juveniles or initial phase individuals, which is consistent with earlier studies concluding that parrotfish are settling into the Mo'orea lagoon and moving to the forereef when reaching larger sizes [42]. The fact that there were a few terminal phase parrotfish observed, suggests that there may be habitats important for these larger-bodied reproductive males such as exposed reef crests and slopes [91], which were not surveyed in the current study. Also, scraping and excavating parrotfish are highly sensitive to top-down control, such as fishing [81], and high fishing pressure in the back-reef lagoon might have reduced the abundance of these larger-sized individuals.

Microhabitat selectivity and fine-scale habitat heterogeneity are important factors structuring reef fish assemblages [92] and there is growing evidence that macroalgal habitats might constitute key habitats for several species or life stages of some species [19,51,63,84,93,94]. Habitats within the Mo'orea lagoon comprising hard coral structure with canopy-forming macroalgae (e.g., *Turbinaria* or *Sargassum* spp.) are likely juvenile habitats that support hotspots of recruitment of certain species of parrotfish. This highlights the critical role that fine-scale soft complexity can play for functionally important fish. However, loss of coral cover (in particular structurally complex branching corals) can induce changes in the fish community, especially causing decline in the abundance of juveniles of certain species [23], and also inhibit settlement of others [95]. Considering the changes in coral community and cover in the Mo'orea lagoon during the past decades, we highlight that the current seascape with the presence of macroalgae and areas of dead coral substrate may favor some species of parrotfish associated with degraded stages of coral reefs (i.e., *C. spilurus*) [96], but may not be attractive for all species.

In conclusion, both seascape configuration (dense/sparse arrangement of hard structure) and the relative presence of macroalgae growing on hard structure were important variables explaining patterns of reef fish abundance and herbivory. However, these relationships differed among functional groups of fish and life-history stages. Information on these habitat-fish interactions are critical for the successful management of coral reef ecosystems, and pinpointing habitat types (e.g., habitat types hosting large numbers of juveniles) that facilitate ecological functions enables their protection.

**Supplementary Materials:** The following are available online at http://www.mdpi.com/1424-2818/12/10/391/s1, Figure S1: Schematic of the complexity categories at different scales and the number of survey points in each, Figure S2: Plots showing variables included in the most parsimonious GAM models for explaining fish abundance (a) total fish (b) juvenile parrotfish and (c) adult parrotfish, Table S1: Abiotic variables derived from point count surveys in the two seascape configuration categories, Table S2: Mean values of focal bommie diameter in the two different seascape configuration categories, Table S3: Mean values ± SE of fish abundance from point count surveys (19.625 m$^2$). Letters in brackets indicate functional group, Table S4: Summary statistics from ANOVA models of the interactive effects of seascape configuration (dense versus sparse bommies), bommie height (short or tall), and the presence/absence of macroalgae on the abundances of all fishes and on herbivorous fishes, Table S5: Summary statistics from ANOVA models of the interactive effects of seascape configuration (dense versus sparse bommies), bommie height (short or tall), and the presence/absence of macroalgae on the abundances of adult and juvenile parrotfishes, Table S6: ANOVA-table of mean distance travelled per min, by *C. spilurus* in different habitat types and size classes.

**Author Contributions:** Conceptualization, C.B., M.E., D.H.C., L.E., J.v.L.; methodology, L.E., M.E., D.H.C., and J.v.L.; formal analysis, M.E. and D.H.C.; investigation, L.E., M.E., D.H.C., and J.v.L.; resources, C.B., C.J.F., C.H.; data curation, M.E. and D.H.C.; writing—original draft preparation, M.E. and D.H.C.; writing—review and editing, C.B., C.H., D.H.C., L.E., M.E., C.J.F., J.v.L., S.W.; visualization, M.E. and D.H.C.; supervision, C.B., C.J.F., C.H., and S.W.; project administration, M.E. and D.H.C.; funding acquisition, C.B., C.J.F., and C.H. All authors have read and agree to the published version of the manuscript.

**Funding:** This research was funded by the Swedish Research Council (Grant numbers 2015-05848, 2015-01257, E0344801).

**Acknowledgments:** We would like to thank the staff at the CRIOBE Mo'orea field station. We are also grateful to Rebecca Fisher for her valuable help with the GAM models and R scripts. We thank the University of South Florida, College of Marine Science for supporting D.H.C. travels and stay at the CRIOBE field station.

**Conflicts of Interest:** The authors declare that there are no conflicts of interest.

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
