# Peer review of "Seascape Configuration and Fine-Scale Habitat Complexity Shape Parrotfish Distribution and Function across a Coral Reef Lagoon"

_diversity, doi:10.3390/d12100391_

Round 1

Reviewer 1 Report

The authors present a potentially interesting work on the relationship between reef fish ecology and habitat diversity at different scales. I find the manuscript well written but i have concerns on the interpretation of statistical analyses on collected data. The authors do not to present the analysis of interactions among the many factors and environmental variables they have analysed, and directly present the significance values of single factors. Factor interactions should be analysed first, as they may strongly affect the overall pattern of data interpretation. I suggest the authors to analyse also factor intractions and reframe their discussion based on the potentially different results. Please also clarify if assumptions for parametric statistics were met for each test, otherwise a non-parametric analogous should be applied.

A minor point: the plots could be made much more immediate if results of statistical analyses among groups are included, so the reader does not have to jump between pictures and tables to understand the overall pattern.

Reviewer 2 Report

Dear Authors,

I found your manuscript interesting and well written but have some minor edits and review request before it can be published, in my opinion!

Please see all comments in the attached annotated PDF and do not hesitate to contact me for any clarifications regarding my comments. I am not a fish ecologist so it might be that several of my thoughts are not educated, please feel free to ignore those.

Good luck,

A reviewer

Reviewer 3 Report

The present manuscript “Seascape configuration and fine-scale habitat complexity shapes parrotfish distribution and function across a coral reef lagoon” represents an interesting contribution in their field. Work is easy to read and presented in a correct form. The main findings and conclusions are clearly separated from the speculative interpretations. Despite this, I have missed some explanations in methodology that could arise some misinterpretations in the main conclusions.

My main concern has been in the UVC protocols and methodology. My first suggestion to the authors is an updated explanation of the sampling period (between 09:00 a.m. and 16:00 p.m. is long and unclear). As far as I know in fishes as in other marine groups (i.e. crustacean decapods) the diel peak activity is important to ensure a correct interpretation/description of a certain community behaviour (i.e. two hours after the sunset and two hours before the sunrise is the maximum peak of activity in some predators and decapod species). In the present work, possible bias could appear between earlier and later surveys. I invite the authors to check the sampling period and test the differences between these factors.

Please see: Bijoux, J. P., Laurent Dagorn, J-C. Gaertner, P. D. Cowley, and J. Robinson. "The influence of natural cycles on coral reef fish movement: implications for underwater visual census (UVC) surveys." Coral Reefs 32, no. 4 (2013): 1135-1140.

Second suggestion is related to the number of samplers (snorkelers) and the number of replicates in each survey. To my knowledge the UCV is the fastest and easiest way to describe fish abundance but it's not exempt from weakness related to a poor number of replicates. I agree with the authors that the point method is the most appropriate methodology. But the high level of tested factors and the amount of biases in the methodology could arise some misinterpretations. I greatly appreciate it, if the number of samplers and number of replicates of each survey was provided.

Specific comments:

1.- Fine-scale structural complexity was estimated on a scale with 5 categories but in figure 2c the number of locations belonging to each category was unbalanced (n0 = 0, n1 = 0, n2 = 5, n3 = 27, n4 = 22, n5 = 32). Which implications had these in the statistical analysis? I appreciate an explanation.

2.- I appreciate the detailed methodologies used to achieve Sea urchins abundance.

3.- Related to fish spatial movement. There are some studies related to the effect of sampler on the behavior of fishes. I suggest that this part of the results was clarified. I recognise that a great effort was made by the authors related to these fields, but I have my concerns about the  influence of the snorkeler on the behavior of fish (1-3 m depth and 2 m between fish and sampler is a short distance to ensure no effects on fish behavior or tortuosity). In my opinion, the snorkeler could operate as a predator. Despite this, I suggest to the authors keep the results and indicate posibles bias related to the sampling.

4.- I found the figures and graphics very illustrative and easy to interpret. My only criticism is related to the colours used in each category. To my knowledge a gradient colour palette is appropriate in a continuous variable but in the categories case I appreciate a great contrast between each group (i.e piscivores and planktivores in figure 3).

Finally I found the discussion well conducted and acertivelly concluded. As I indicate in the beginning, my main concerts are in some methodological aspects. All the aspects mentioned in the methodology have a direct impact in the discussed results, I appreciate some comments on the possible biases and impact of the sampling strategies on the results.

Round 2

Reviewer 1 Report

The authors have convincingly addressed all my previous comments.

Author Response

Thank you for your input and reviews. We are glad to hear the edits to the manuscript were well received.

Reviewer 3 Report

General comment (Round 2)

Congratulations to the authors. I have found the main suggestions and concerns covered by the authors. Some minor comments were not addressed but it was not a mandatory.

Reviewer 3:

The present manuscript “Seascape configuration and fine-scale habitat complexity shapes parrotfish distribution and function across a coral reef lagoon” represents an interesting contribution in their field. Work is easy to read and presented in a correct form. The main findings and conclusions are clearly separated from the speculative interpretations. Despite this, I have missed some explanations in methodology that could arise some misinterpretations in the main conclusions.

My main concern has been in the UVC protocols and methodology. My first suggestion to the authors is an updated explanation of the sampling period (between 09:00 a.m. and 16:00 p.m. is long and unclear). As far as I know in fishes as in other marine groups (i.e. crustacean decapods) the diel peak activity is important to ensure a correct interpretation/description of a certain community behaviour (i.e. two hours after the sunset and two hours before the sunrise is the maximum peak of activity in some predators and decapod species). In the present work, possible bias could appear between earlier and later surveys. I invite the authors to check the sampling period and test the differences between these factors.

Please see: Bijoux, J. P., Laurent Dagorn, J-C. Gaertner, P. D. Cowley, and J. Robinson. "The influence of natural cycles on coral reef fish movement: implications for underwater visual census (UVC) surveys." Coral Reefs 32, no. 4 (2013): 1135-1140.

Response: thank you for your concerns on the analyses, we conducted the surveys between 9-16 to avoid any behavioral differences associated with sunrise or sunset. Additionally, we chose that time of surveying because that is when herbivorous fish are mostly active. We have added more details in the methodology to make this point clear on lines 111-115:

“Field surveys were conducted along the northern shore of Mo’orea (Fig. 1). All surveys were performed within the lagoon (the back-reef habitat) at 1-3 m depth during October 2017. All surveys were conducted between 09:00-16:00 to avoid any behavioral differences associated with sunrise or sunset [47]. This survey period also covers the most active time for herbivorous fish, which were the focus of this study [48].”

We have also acknowledged in the discussion that other functional groups may have different relationships with hard substrate during dark hours due to predator/prey diel variation on lines 433-435:

 “Similarly, the role of hard structure during night time may differ in other fish functional groups such as invertivores as some may follow invertebrate diel peak activity and/or move to other habitats for feeding [47,83].”  

Reviewer Round 2: Dear authors. I accept that the differences between early and later UVC haven't been tested. As I said it was a suggestion. The explanation of sampling protocol has been conducted correctly.

Second suggestion is related to the number of samplers (snorkelers) and the number of replicates in each survey. To my knowledge the UCV is the fastest and easiest way to describe fish abundance but it's not exempt from weakness related to a poor number of replicates. I agree with the authors that the point method is the most appropriate methodology. But the high level of tested factors and the amount of biases in the methodology could arise some misinterpretations. I greatly appreciate it, if the number of samplers and number of replicates of each survey was provided.

Response: We have added more details to the methods in regards to the UVCs on lines 138-141. Also, figure 2 and figure 4 have the number of replicates per category specified.

“To minimize any variation related to the number of observers, all Underwater Visual Census (UVC) were performed by ME and DHC. We practiced estimating fish lengths prior to the UVCs to minimize size estimate errors and to calibrate both observers [28]. Forty-two UVC were carried out in dense habitats and 44 in sparse habitats (Fig. 2).”

Reviewer Round 2: Personally, I consider a single data as the mean value of two UVC at each single sampling point (conductet by two scuba divers). These is due to the UVC data were characterised by high variability, low precision and low power. Authors describe data as single UVC. I have accepted these aproach due to the hight number of UVC conducted in each area. 

see: Samoilys, Melita A., and Gary Carlos. "Determining methods of underwater visual census for estimating the abundance of coral reef fishes." Environmental Biology of Fishes 57.3 (2000): 289-304.

Specific comments:

1.- Fine-scale structural complexity was estimated on a scale with 5 categories but in figure 2c the number of locations belonging to each category was unbalanced (n0 = 0, n1 = 0, n2 = 5, n3 = 27, n4 = 22, n5 = 32). Which implications had these in the statistical analysis? I appreciate an explanation.

Response: This occurred because we did not specifically choose locations covering all the different fine-scale complexity categories. Our survey locations were based on: 1) configuration of bommies (dense or sparse), 2) presence/absence of macroalgae, and 3) short or tall bommie size. After each sampling location was chosen based on these criteria, we further measured finer-scale habitat complexity categories in the scale 0-5, thus producing an unbalanced number of sampling points for each finer-scale complexity category. Ideally, we would have had equal numbers for everything we measured, but this was not the case and reflects the natural variation of fine-scale complexity in the seascape. We have added a sentence in the methods to clarify this point in lines 115-116:

“Our survey locations were selected based on three criteria: 1) configuration of coral bommies, 2) presence/absence of macroalgae, 3) height of bommie.”

and have acknowledged this potential effect in the discussion on lines 427-431:

“Surprisingly we could not detect an effect of hard fine-scale complexity (scale 0-5) on fish abundance in the UVCs, which could have been influenced by the low number of replicates in the different categories. This was further supported by the results from our rugosity calculations using photogrammetry, where rugosity did have an effect on fish abundance.”

Reviewer Round 2: Point accepted

2.- I appreciate the detailed methodologies used to achieve Sea urchins abundance.

Response: thank you for your suggestion, we have provided more details in the methodology on lines 158-159:

“Within the surveyed point census area, the sea urchin abundance was also noted.”

Reviewer Round 2: Point accepted

3.- Related to fish spatial movement. There are some studies related to the effect of sampler on the behavior of fishes. I suggest that this part of the results was clarified. I recognise that a great effort was made by the authors related to these fields, but I have my concerns about the influence of the snorkeler on the behavior of fish (1-3 m depth and 2 m between fish and sampler is a short distance to ensure no effects on fish behavior or tortuosity). In my opinion, the snorkeler could operate as a predator. Despite this, I suggest to the authors keep the results and indicate posibles bias related to the sampling.

Response: we agree in that observers’ effects are hard to minimize and we have therefore acknowledged this in the manuscript. However, we paid attention to any indication of disturbance caused by the observer such as abrupt interruption of feeding or burst of fast swimming in opposite direction. If a disturbance was assessed, the trial was terminated. Additionally, we assumed that any potential observer effects occurred consistently across our fish following trials. We have re-emphasized this in our manuscript in lines 173-176:

“If fish behavior appeared to be influenced by the snorkeler (e.g., abrupt interruption of feeding, sudden fast swimming in opposite direction, unwanted attention towards the snorkeler), the session was terminated and the data excluded from analyses. In case of unnoticed disturbance, we assumed that these occurred similarly across our trials.”

Reviewer Round 2: I have found the explanation of the authors acceptable. Observations aren't free of interactions in any case, but it could be asummed.

4.- I found the figures and graphics very illustrative and easy to interpret. My only criticism is related to the colours used in each category. To my knowledge a gradient colour palette is appropriate in a continuous variable but in the categories case I appreciate a great contrast between each group (i.e piscivores and planktivores in figure 3).

Response: great suggestion. We have modified the figures to more clearly display the results. Edited figure 3 is found in lines 290-295. Edited figure 4 is found on lines 307-308, and edited figure 5 on lines 314-320.

Reviewer Round 2: Point accepted. I have founded more easy to read now.

Finally I found the discussion well conducted and acertivelly concluded. As I indicate in the beginning, my main concerts are in some methodological aspects. All the aspects mentioned in the methodology have a direct impact in the discussed results, I appreciate some comments on the possible biases and impact of the sampling strategies on the results.

Response: thank you for your suggestions. We have made sure we address and/or acknowledge your suggestions in the revised manuscript. For example, we acknowledged your concern about unbalanced replicates among categories in the discussion in lines 427-431:

“Surprisingly we could not detect an effect of hard fine-scale complexity (scale 0-5) on fish abundance in the UVCs, which could have been influenced by the low number of replicates in the different categories. This was further supported by the results from our rugosity calculations using photogrammetry, where rugosity did have an effect on fish abundance.”

We also acknowledged the diel activity of prey for other functional group of fish in the discussion in lines 433-435:

“Similarly, the role of hard structure during night time may differ in other fish functional groups such as invertivores as some may follow invertebrate diel peak activity and/or move to other habitats for feeding [47,83].”

Reviewer Round 2: point accepted

Conglaturations for the efford

Author Response

(The authors gave the same response as above.)
